# The Metamorphosis (of RAM³S)

Ilaria Bartolini *,† and Marco Patella †

Department of Computer Science and Engineering (DISI), Alma Mater Studiorum, University of Bologna, Viale del Risorgimento, 2, 40136 Bologna, Italy; marco.patella@unibo.it
* Correspondence: ilaria.bartolini@unibo.it
† These authors contributed equally to this work.

**Abstract:** The real-time analysis of Big Data streams is a terrific resource for transforming data into value. For this, Big Data technologies for smart processing of massive data streams are available, but the facilities they offer are often too raw to be effectively exploited by analysts. RAM³S (Real-time Analysis of Massive MultiMedia Streams) is a framework that acts as a middleware software layer between multimedia stream analysis techniques and Big Data streaming platforms, so as to facilitate the implementation of the former on top of the latter. RAM³S has been proven helpful in simplifying the deployment of non-parallel techniques to streaming platforms, such as Apache Storm or Apache Flink. In this paper, we show how RAM³S has been updated to incorporate novel stream processing platforms, such as Apache Samza, and to be able to communicate with different message brokers, such as Apache Kafka. Abstracting from the message broker also provides us with the ability to pipeline several RAM³S instances that can, therefore, perform different processing tasks. This represents a richer model for stream analysis with respect to the one already available in the original RAM³S version. The generality of this new RAM³S version is demonstrated through experiments conducted on three different multimedia applications, proving that RAM³S is a formidable asset for enabling efficient and effective Data Mining and Machine Learning on multimedia data streams.

**Keywords:** stream processing; real-time analysis; big data; multimedia data streams

*One morning, as the authors were waking up from anxious dreams, they discovered that RAM³S had to be changed into an up-to-date stream processing framework.*

## 1. Introduction

Multimedia (MM) data have been used in a plethora of applications for at least three decades. The wide availability of such data is facilitated by the presence of low-cost production (cameras, sensors, etc.) and storage devices. Moreover, often MM data comes in streams, i.e., sequences of MM objects of the same type that are received from a producer.

The value that can be extracted from MM streams is dramatic, but so are challenges posed on computing power and analytical intelligence [1]. In particular, real-time analysis requires that data streams are processed with high throughput and limited latency, so as to exploit data freshness to respond and make decisions quickly.

In this paper, we describe the metamorphosis of RAM³S [2,3], a framework that we developed to integrate Big Data management platforms, so as to allow researchers to easily implement real-time complex analyses of massive MM streams exploiting a distributed computing environment (RAM³S stands for Real-time Analysis of Massive MultiMedia Streams).

The use of RAM³S allows researchers to easily adapt complex MM stream analysis techniques (initially conceived for a centralized system) to a distributed computing scenario, so as to effectively *scale out* them. Indeed, Big Data platforms offer a number of services which are crucial for the management and analysis of large quantities of information, in order to provide evidence-based decision making in many aspects of human activities. However, the use of such platforms is made difficult for analysts by the fact that their

major focus is on issues of fault-tolerance, synchronization, increased parallelism, and so on, instead of providing an easy-to-use interface to programmers.

The idea behind RAM$^3$S was not to create yet another streaming processing engine, rather to give MM analysts an easy-to-use interface to existing Big Data streaming platforms. RAM$^3$S thus acts as a bridge to integrate two complex technologies, one applicative (the multimedia stream analysis) and one architectural (the Big Data platform), allowing the seamless implementation of the former onto the latter. In this way, for example, a developer could compare the performance of different streaming engines before committing to a specific one for her applications. The fields of application of such technologies are innumerable, from the security of citizens [1], to recommender systems [4], and sentiment analysis [5], to cite a few.

Although the generality of RAM$^3$S has been demonstrated before, its applicability to the ever-changing world of stream analysis was limited by two main issues:

- First, in its original incarnation [2], RAM$^3$S exploited RabbitMQ to feed stream data into the system, and no other message broker was considered as an alternative;
- Second, the streaming processing model was over-simplistic, consisting of just a single node where each MM objects was analyzed.

The contributions of the new, metamorphosed version of RAM$^3$S we present here are:

1. The RAM$^3$S interface to input (and output) streams has been generalized so as to encompass the use of different message brokers (the current version includes the interface to three different message brokers as opposed to the single one originally available);
2. Decoupling RAM$^3$S from the message broker also allows cascading different RAM$^3$S instances, since the output of a single instance can now be used as input to another one. This provides a recursive, and richer, model for the processing of MM streams;
3. As an additional feature, we also included in RAM$^3$S another popular Big Data streaming platform, namely Apache Samza, in addition to the ones already present (Spark Streaming, Storm, and Flink);
4. Finally, thanks to the wide availability of alternatives provided by this new version of RAM$^3$S, we are able to experimentally compare performances of different message brokers and stream processing engines on three different real-world applications, demonstrating the handiness of RAM$^3$S as a platform for testing purposes.

We believe that this new version of RAM$^3$S represents a formidable asset for enabling efficient and effective Data Mining and Machine Learning on MM data streams.

### 1.1. Running Example: Face Recognition

The first use case we developed with RAM$^3$S was the automatic recognition of known faces in videos [2]. In this example, which we will use throughout the paper to illustrate the various RAM$^3$S components, a number of incoming videos are fed to the system, which should detect and identify faces contained in each frame. An example of use of such technique is the detection of criminals in videos obtained through cameras disseminated in the territory (such as airports, metro stations, public buildings, and so on) For this, each video is first streamed as a sequence of frames and each incoming frame is examined to:

1. Check whether it contains a face;
2. Compare the (possibly) discovered face against a number of "known" faces, to retrieve the known face most similar to the input face;
3. If the similarity between the discovered face and its most similar known face is sufficiently high, the face is considered as correctly recognized, otherwise it is regarded as an unknown face.

For this use case, we exploited the well-known Viola–Jones algorithm [6] for face detection, while comparisons of "faces" was performed using a technique based on principal component analysis using eigenfaces [7]. For the purpose of suspect identification, whenever a discovered face is sufficiently similar to one of the faces in the knowledge base, an alarm is raised (see Figure 1a).

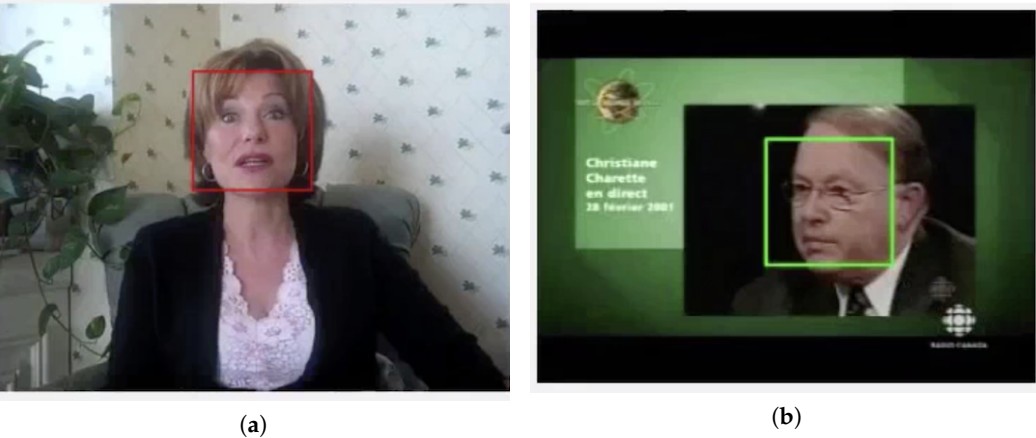

(**a**)     (**b**)

**Figure 1.** Face recognition use case: known (**a**) and unknown (**b**) face.

*1.2. Paper Outline*

After describing the context that led us to devise RAM$^3$S (Section 2), we provide a high-level view of the framework and its model (Section 3). In Section 4 we describe Apache Samza, while in Section 5 we show how the latter stream processing platform has been included in this novel RAM$^3$S implementation. We then report a series of experiments we conducted in our datalab over a number of application where RAM$^3$S was effectively exploited to build useful Web-based services in the fields of citizens security, traffic control, and product recommendation (Section 6). Finally, we conclude, drawing possible future research directions (Section 7).

## 2. In the Beginning

Our experience with Big Data streaming analysis platforms derives from a private-financed project focused on the analysis of multimedia streams for security purposes. The private company already had an experience in Big Data *batch processing*, through the long term use of tools such as Hadoop (http://hadoop.apache.org, accessed on 3 December 2021) and Spark. (http://spark.apache.org, accessed on 3 December 2021). However, it was clear that the use of such technologies was inappropriate for a number of security tasks, such as face detection for the automatic identification of "suspect" people [7], recognition of suspicious behavior from videos [8], human actions [9] or gesture [10], audio events [11], and so on. Indeed, in such *online* applications, data have to be analyzed as soon as these are available, so as to exploit their "freshness". The storing of incoming data is, thus, usually unnecessary and the efficiency of the system depends on the amount of data processed, keeping low latency, at the second, or even millisecond, level.

To deal with this (at the time) novel *stream processing* paradigm other Big Data platforms were introduced, among which Storm, (http://storm.apache.org, accessed on 3 December 2021) Flink, (http://flink.apache.org, accessed on 3 December 2021) and the streaming version of Spark. Abstracting by specificities of each Big Data streaming platform (see [12,13] for details on Spark and Flink, respectively), the following common characteristics can be discovered (see also [2] for a more detailed comparison of the streaming processing platforms considered in RAM$^3$S):

- Some nodes in the architecture are in charge of receiving the input data stream, thus containing the data acquisition logic (these are called Receivers in Spark, Spouts in Storm, and Producers in Flink);
- Other nodes perform the actual data processing (these are called Drivers in Spark, Bolts in Storm, and Task Nodes in Flink);
- Finally, data sources and data processing nodes are connected to realize the data processing architecture, which takes the form of a Directed Acyclic Graph, where arcs represent the data flow between nodes (in Storm, such architecture is termed Topology).

We were, therefore, challenged to implement a number of security-related use-cases on top of such stream processing Big Data platforms, with the goal of comparing them by way of several performance KPI, such as throughput, scalability, latency, etc. This required to re-implement every multimedia stream analysis application on top of each platform, leading to huge code replication and other inefficiencies. We were therefore led to consider realizing a middleware software framework to allow users to:

1. Avoid having to deal with details of each specific Big Data streaming platform (such as how fault-tolerance is achieved, how stream data are stored, etc.); and
2. Easily extend already available (centralized) software to a *scaled out* solution.

This way, we strove to bridge the technological gap between facilities provided by Big Data analysis platforms and advanced applications (for which transition to a distributed computing scenario might not be straightforward).

### 2.1. The Message Broker

A message broker is a software module that allows applications, systems, and services to communicate with each other, translating a message from the sender's formal communication protocol to the receiver's one. This allows interdependent services to "talk" directly with each other, even if they have been written using different languages or implemented on different platforms. Message brokers act as intermediaries between different applications, so senders can send messages without knowing where the recipients are, whether they are active or not or how many of them are present. This clearly facilitates the decoupling of processes and services within systems. The task of a message broker is obviously not only to pass data, as it also deals with aspects related to security, priority, and orderly forwarding of messages.

Typically, the message broker is instantiated in a server process communicating with both the sender and the receiver using their respective communication protocols, thus providing status control and tracking of all customers who are involved in the communication, in order to avoid that individual applications should implement the aforementioned functionalities and complex message delivery.

Generally speaking, the entities that interact in the Message Broker context are:

- One or more Producers, i.e., processes generating messages;
- One or more Consumers, i.e., processes dealing with the handling of messages;
- The Message Broker himself, storing messages coming from Producers and forwarding them to the Consumers.

The nature of middleware allows a message broker to decouple heterogeneous systems which, with other configurations, would be strongly coupled; a message broker can validate, store, direct, and deliver messages to the correct destinations, in fact they are typically used as intermediaries between applications. This implies that producers can deliver messages to brokers without even knowing which consumers will process them. To provide a reliable message storage system and a guarantee of message delivery, brokers typically rely on a structure called a message queue that stores and sorts messages until consumers have processed them.

Message brokers commonly use an asynchronous model of message communication: This prevents the loss of important data and allows systems to continue operating even in presence of intermittent connectivity or latency problems, as typical in public networks. Asynchronous messaging ensures that messages will be delivered once (and only once) in the correct order relative to other messages.

#### 2.1.1. Communication Models

Message brokers typically offer two basic message exchange models:

Point-to-Point model: This model considers a one-to-one relationship between the sender of the message and the recipient. Each message in the queue is sent to only one consumer and is consumed exactly once. Once the message is consumed, the recipient

sends a confirmation signal (an *acknowledgement*) to the broker who can then delete the message from its queue. Point-to-point messaging is required when a message only needs to be processed once. The model also provides for the possibility of having multiple consumers connected to a queue, but a specific message will always be consumed by at most one of them. Examples of cases for this messaging model include payroll and financial transaction processing: In these systems, both the sender and the recipient need to be guaranteed that each payment will be sent only once.

Publish/Subscribe model: In this messaging model, often referred to as "pub/sub", the producer publishes the message on a topic and multiple consumers subscribe to the topics they want to receive messages from. All messages published on a topic are distributed to all applications subscribed to it. This is a broadcast-style distribution method, where there is a one-to-many relationship between the publisher of the message and its subscribers. For example, if an airline were to issue updates on landing times or the delayed status of its flights, multiple receivers could be interested in such information: The ground crews who maintain and refuel the aircraft, baggage handlers, flight attendants, and pilots preparing for the next trip, and travelers. Unlike the Point-to-Point model, which groups messages until they are retrieved by consumers, topics immediately transfer messages to all their subscribers. All client processes subscribing to a given topic will then receive every transmitted message, unless they have set up a message filtering policy.

The first version of RAM$^3$S only considered RabitMQ as message broker, while the current version also supports ActiveMQ and Kafka. All currently supported message brokers are described in the following sections.

### 2.1.2. RabbitMQ

RabbitMQ (https://www.rabbitmq.com/, accessed on 3 December 2021) is an open-source message broker that implements the Advanced Message Queuing Protocol (AMQP) 0.9.1. The RabbitMQ server is written in Erlang and is based on the Open Telecom Platform (OTP) framework for managing clustering and failover. It is considered to be one of the most popular open-source message brokers used worldwide by small start-ups up to large enterprises. Its popularity stems from its ease of installation and also from the broad support provided for programming languages and operating systems.

RabbitMQ was originally launched in 2007 by the collaboration between Lshift and CohesiveFT as a complete open source implementation of the AMQP protocol, the emerging standard for high performance enterprise messaging. In April 2010, RabbitMQ was acquired by SpringSource, a division of VMware, which added the messaging system to its suite of technologies, to reduce the complexity associated with developing, deploying, and managing enterprise applications. The project then became part of Pivotal Software in 2013.

### 2.1.3. ActiveMQ

ActiveMQ (http://activemq.apache.org/, accessed on 3 December 2021) is an open source messaging software, written completely in Java, which allows the creation of a multi-protocol, highly available, and highly scalable messaging system. It can therefore serve as the backbone for a messaging-based distributed application architecture. ActiveMQ is a Java Message Service (JMS) provider, which means that it implements the functionalities specified in the JMS API. Client applications, producers, and consumers can, therefore, use the JMS API to send and receive messages. However, ActiveMQ also offers support for other protocols such as AMQP, MQTT, and STOMP.

The ActiveMQ project was originally created by LogicBlaze in 2004, as an open source message broker, hosted by CodeHaus. The ActiveMQ code and branding were donated to the Apache Software Foundation in 2007, where the founders continued to develop the codebase with the extended Apache community.



### 2.1.4. Kafka

Apache Kafka (https://kafka.apache.org/, accessed on 3 December 2021) is an open source distributed event streaming platform developed by the Apache Software Foundation and used by thousands of companies for high-performance data pipelines, streaming analytics, data integration, and mission-critical applications. Written in Scala and in Java, the project aims to provide a unified, high-performance, low-latency platform for managing real-time data feeds. Apache Kafka can connect to external systems (for data import/export) via Kafka Connect and provides Kafka Streams, a Java stream processing library.

Kafka combines three key features to enable implementing use cases for end-to-end event streaming with a single solution in a distributed, highly scalable, elastic, fault tolerant, and secure way:

- Writing and reading event streams, including continuous import/export of data from other systems;
- Archiving streams of events in a durable and reliable way;
- Processing flows of events as they occur or retrospectively.

Kafka was originally developed by LinkedIn and became open source in 2011.

## 3. Let There Be RAM$^3$S

The first step towards the definition of a framework for the analysis of massive multimedia streams came from the realization that, in all the use cases we considered, incoming data streams were processed as illustrated in Figure 2 (we note that the simple model presented here has been appropriately extended in the current version of RAM$^3$S to encompass other, more complex, types of analyses):

1. First, the input data stream is transformed in a sequence of individual MM objects (such as images or small video/audio sequences);
2. Then, each single MM object is analyzed to derive a result (e.g., an alarm is raised if a suspect person is recognized in an image or a particular audio event is detected).

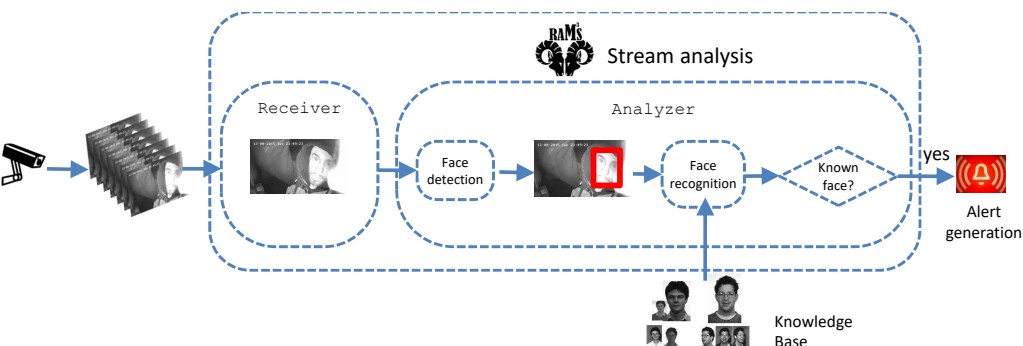

**Figure 2.** Analysis of a multimedia data stream with RAM$^3$S.

In this case, the topology of nodes in the architecture is extremely simple, being composed of several data sources which are connected to a network of data processing nodes, each of which is able to deliver its result by analyzing a single MM object. The underlying architecture is therefore extremely standardized and is independent of the actual application and/or the particular MM object/stream. In this way, each application can be implemented on top of each Big Data platform by simply specifying how:

1. Each MM stream can be broken up into a sequence of individual MM objects; and
2. Every single MM object can be analyzed to produce (a part of) the result.

The RAM³S framework thus provides a common interface to specify the two above specified behaviors, introducing the `Receiver` and the `Analyzer` abstract components. Both interfaces are extremely simple:

- For the `Receiver` component, only the `start()` and the `stop()` methods should be defined, specifying how the data stream acquisition is initiated and terminated; the output of a `Receiver` is a sequence of multimedia object instances;
- On the other hand, the `Analyzer` only includes a constructor and a single `analyze()` method, the latter taking a single multimedia object instance as input, producing a boolean value in output.

When cascaded, the `Receiver` and the `Analyzer` components implement the generic functionalities of a stream analyzer, where the continuous flow of input data produces a sequence of booleans, indicating activations/de-activations of the alarm.

Clearly, the RAM³S framework also includes other classes, which are not detailed here, that implement the connection to the underlying Big Data platforms: All issues specific to distributed data processing are dealt with within these classes, so that the intricacies of distributed computing (such as fault-tolerance, isolation, and stateful processing) are hidden to the user, who can focus on how to adapt her data stream analysis technique to the `Receiver` and the `Analyzer` interfaces.

## 4. Introducing Samza

Big Data streaming platforms that were included in the original RAM³S incarnation were Spark Streaming, Storm, and Flink [2]. To further expand features included in RAM³S, we considered to add Apache Samza to the supported streaming processing platforms. Apache Samza (https://samza.apache.org/, accessed on 3 December 2021) [14] is a distributed framework for stream processing originally developed by LinkedIn in 2013, which in 2014 became an Apache top-level project.

The advantage introduced by Samza is the possibility of using a single callback-based API to describe the application logic independently of the data source, which can be both batch and streaming. Furthermore, the framework allows to process and to transform data obtained from various sources: Samza offers a built-in integration with Apache Kafka, AWS Kinesis, Apache Hadoop, Azure EventHubs, and ElasticSearch. Another point in favor of Samza is that it can be used as a normal library integrated into a Java/Scala project eliminating the need to use a separate cluster for stream processing: It becomes a real library embedded in the application. This mode of use is called stand-alone mode and allows one to have control over the entire life cycle of the application. The stand-alone mode also gives the possibility to increase the capacity of the application by creating multiple instances; all such instances will coordinate with each other to be able to equally balance the work: If an instance should fail for any reason, the task assigned to it would be reassigned to another instance. Coordination between the instances is carried out through Zookeeper but, since the coordination logic is pluggable, it can also be dealt with other frameworks.

The other deployment mode supported by Samza is execution on Apache YARN: In this modality, a Samza application is sent to YARN to be then executed on its cluster. Thanks to this integration with YARN, isolation, multi-tenancy, resource management, and application deployment are provided; the number of containers needed, the number of cores, and the memory to be reserved for each container can also be specified. In addition, Samza works with YARN to provide the necessary resources for the application at hand and run it in a cluster of machines and to automatically restart failing application instances. When multiple applications share the same YARN cluster, they must be isolated from each other: Samza works with YARN to set CPU usage and memory occupation limits. As a result, any application that uses more than it is allowed is automatically terminated, conferring multi-tenancy on the system. Each deployment model brings its benefits in order to give flexibility in choosing which one to adopt.

Finally, it is important to underline the at-least-once semantics adopted by Samza which ensures that every message coming from an input stream is processed at least once; this model ensures that no losses occur in the event of failures, making Samza a fault-tolerant choice for applications. Moreover, Samza is the only system that provides support for both deployment systems: Some systems, such as Kafka, only support the library model, while others, such as Flink or Spark streaming, only provide the framework model for stream processing.

### 4.1. Samza Streams, Partitions, and Jobs

Samza processes the data as organized in *stream*s. In particular, a Samza stream is a collection of immutable messages, typically organized by categories, where each message belonging to the stream is modeled as a <key, value> pair. A stream can have multiple producers writing its data and as many consumers reading its data. In particular, the data can come from infinite sources (e.g., Kafka topics) or from finite sources (e.g., a set of HDFS files). A stream is then separated into multiple *partitions* to make its manipulation more scalable: Each partition is, therefore, an ordered sequence of records. When a message is written to the stream, it is stored in one of its partitions and identified by a specific offset. Samza allows to adopt various systems to realize stream abstraction; for example, a topic coming from Kafka or even a sequence of updates to the tables of a database. A *job* is code that consumes a set of input streams. To make the system throughput as scalable as possible, jobs are separated into smaller units called *tasks*.

Samza provides scalability to applications by logically dividing them into multiple tasks: A task is the logical unit of parallelism of the application. Each task manipulates data from one or more partitions for each input stream. Since there is no specific order between partitions, tasks can operate on them independently of each other. In the event that a task is running on a failed machine, it is restarted on another machine, always processing the same partition. Since there is no order among the messages belonging to various partitions, the tasks can process entirely independent of each other without having to share state.

Just as a task is the logical unit of parallelism of the application, a *container* is its physical unit: Each worker can be thought of as a JVM that executes one or more tasks. An application that makes use of Samza typically has multiple containers distributed among the various hosts. Each application also includes a Coordinator that manages the assignment of tasks between the various containers. The coordinator also monitors the life cycle of each container and redistributes the tasks among the remaining ones in case of failure. The coordinator itself is pluggable, making Samza able to support various deployment solutions.

### 4.2. Samza APIs

Samza provides two different APIs that can be exploited to build applications: High-level or low-level APIs. In both cases it is possible to describe the application logic in its entirety, with the only difference that with high-level APIs the application processes message streams, while with low-level APIs single messages are the processing unit. Each Samza task receives data from an input stream (possibly split in more than one partition), process such data, and emits results to an output stream (in turn, possibly split in multiple partitions).

When using high-level APIs, the stream processing application is represented as a Directional Acyclic Graph (DAG) operating on streams (each implementing the `MessageStream`
`<M>` interface, where `M` is the type of message contained in the stream). To create an application using high-level APIs, the `StreamApplication` interface has to be implemented, by specifying the `describe()` method, where the (input and output) streams are described, together with the operations to be performed on each message of type `M`.

On the other hand, the use of low-level APIs allows to specify operations on a single message in an object representing a real task (logical unit of parallelism). For this, it is sufficient to implement the `TaskApplication` interface, again by specifying the `describe()`

method, to indicate that each message will be processed by another object, implementing either the `StreamTask` or the `AsyncStreamTask` interfaces, representing, respectively, synchronous and asynchronous tasks. To implement such interfaces, one needs to specify the `process(M)` method, describing how the message `M` has to be processed (the asynchronous interface also requires a callback argument, which is a function that will be invoked when processing of a message is terminated). The advantages of asynchronous communication are clear: Each task in the graph is allowed to process data at its convenience, while synchronous communication is necessary, for example, when a timely response is needed.

### 4.3. Samza and Kafka

The close connection between Samza and Kafka may seem restrictive; however, it gives the system some unique characteristics, not commonly found in others systems for stream processing, somehow mirroring the way MapReduce relies on HDFS. For example, Kafka already provides replicated data storage and can be accessed with low latency. Furthermore, it is also possible to store intermediate results thanks to Kafka and then process them in downstream states. The strong relationship between Samza and Kafka allows the processing steps to be loosely coupled with each other. An arbitrary number of subscribers can be added to the output for each step without affecting the processing of the other subscribers. This can be very useful for organizations when multiple teams need to access similar data: Teams can become subscribers to topics created by other teams without causing stress on the system infrastructure. Writing directly to Kafka also eliminates back-pressure problems, i.e., when the load peaks of a data flow are greater than the components' ability to process in real-time, thus potentially introducing stalls and data losses. Being designed to keep data for certain periods of time, thanks to Kafka the components can process it according to their abilities and start over without causing loss of results. As a result, Samza is a good choice for stream processing and, furthermore, it greatly simplifies stream processing by offering low latency performance.

## 5. The New RAM$^3$S

In this Section we show how RAM$^3$S has been extended to encompass (i) the use of different message brokers and (ii) the Apache Samza streaming platform.

### 5.1. Generalizing the Message Broker

The first extension needed for RAM$^3$S is the ability to specify the message broker used to read the input stream and to write the output stream: The original version of RAM$^3$S was hard-coded to use RabbitMQ. Decoupling RAM$^3$S from the message broker avoids binding the underlying frameworks to a specific choice of message broker, so that a factory (`MessageBrokerFactory`) can be used to obtain a particular message broker that will be automatically used by the frameworks that manage the stream processing. For this, two interfaces were added to RAM$^3$S, so as to manage separately the input and output streams: The `Reader` is directly connected to the RAM$^3$S `Receiver` while the `Writer` is used to store results obtained from the `analyze()` method of the `Analyzer` (the output of the Analyzer::analyze() method has also been modified to return a MM object instance rather than a boolean value). Figure 3 displays an UML diagram of the main interfaces created in RAM$^3$S. The `Receiver` (respectively, the `Analyzer`) uses the `MessageBrokerFactory` to obtain a `Reader` to read MMObjects from (resp., a `Writer` to write results to).

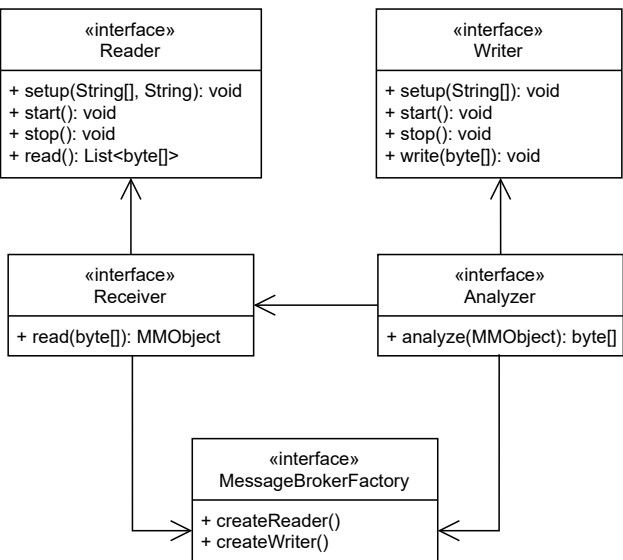

**Figure 3.** UML diagram of the main RAM³S interfaces.

Four methods were included in the `Reader` (resp. `Writer`) interface:

- The constructor offers the possibility to configure the message broker through an array of strings, typically obtained from the `Receiver` (the `Writer` constructor has an additional parameter to possibly specify a name for the output stream).
- The `start()` and `stop()` methods are included in case a specific message broker, implementing the interface, requires some startup/shutdown operation.
- The `read()` (resp. `write()`) method reads (resp. writes) a byte array from/to the stream.

The use of a specific message broker only requires to exploit the particular `Message BrokerFactory` interface provided by RAM³S, so as to obtain the `Reader`/`Writer` objects for the chosen message broker (see Figure 4 for the case of Kafka).

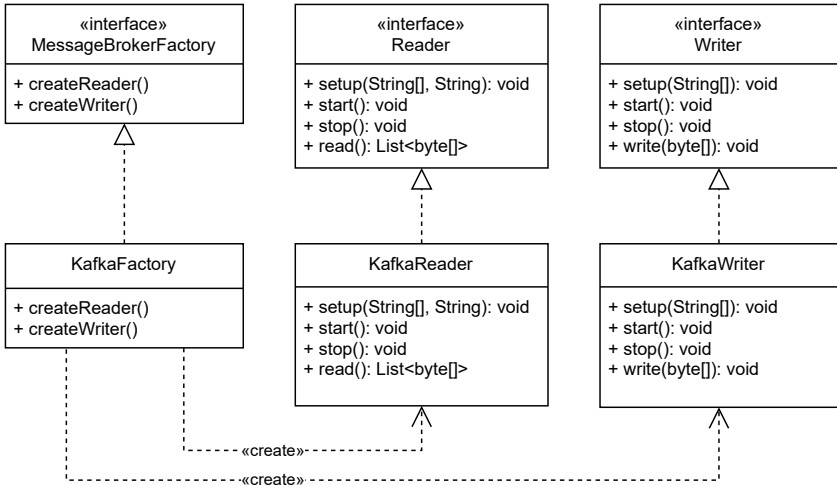

**Figure 4.** UML diagram of the message broker factory for the Kafka case.

The creation of a particular application is obtained by implementing the `Application Factory` interface to produce the specific `Receiver`/`Analyzer` objects. In the `Receiver`, the `read()` method specifies how a sequence of bytes can be used to create a particular `MMObject`, while the `analyze()` method of the `Analyzer` performs the analysis of a single `MMObject`, producing the result in form of a sequence of bytes that can be written on the `Writer`. Figure 5 shows the UML diagram for the face recognition application.

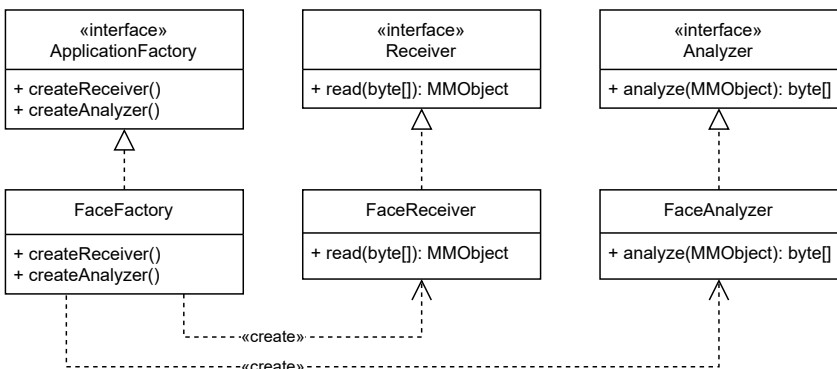

**Figure 5.** UML diagram of the application factory for the face recognition case.

By introducing the `Reader` and `Writer` abstractions, we also allow the new version of RAM³S to be recursively cascaded so as to perform more complex operations on the input stream. Suppose that the first operation performed on the input MM stream produces a new stream of MM objects; this output stream could be redirected as input to another RAM³S instance, analyzing such an MM stream to produce the result: For example, the first input stream could be a stream of documents that are first analyzed to extract images they contain. Such images are then fed (as a new MM stream) to a second RAM³S instance that performs some, say, classification task. This was clearly not possible with the original RAM³S, where all computations were relegated on a single node (where the `analyze()` method was run).

As another example, consider the use case originally presented in RAM³S, concerning the recognition of known faces from incoming videos for security purposes (see also Section 1.1). In this case, for each image in the incoming stream, the RAM³S `Analyzer` should (i) check whether it contains at least a face and (ii) compare all faces in the image to the "known" ones. As also acknowledged in [2], the first task is the most computationally expensive, mostly because, in the considered domain, detection should be particularly accurate so as to avoid the mis-detection of a face. Moreover, while some images may not contain any face, other ones might contain many of them, thus making the recognition task very unbalanced (some images can produce a high latency, while others are processed very quickly). Splitting the two tasks of face detection and face recognition into two cascaded RAM³S instances (see Figure 6) could bring two advantages:

1. First, the second task of face recognition could be better parallelized in the second RAM³S instance, since every `Analyzer` only has to process a single image, thus the overall latency is likely to be reduced;
2. Second, the two instances could be run on different processing nodes, so that a larger number of nodes (or nodes with higher processing power) could be allocated for the most time-requesting task of face detection.

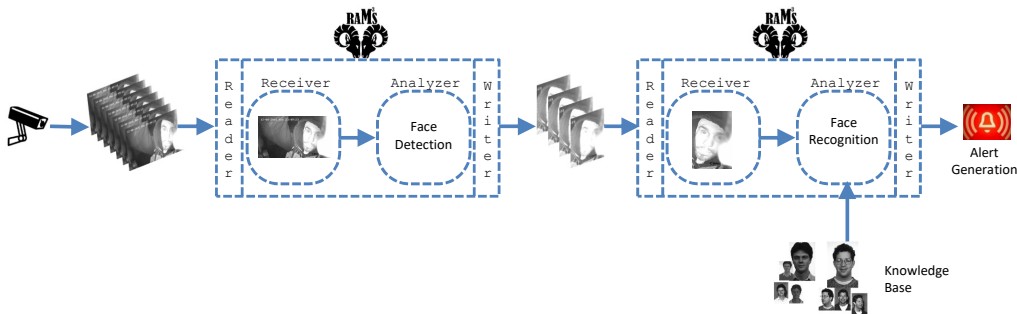

**Figure 6.** Cascading two RAM³S instances.

Currently, the communication model of RAM³S components is the Point-to-Point model, thus each `Writer` can only be connected to a single `Reader`. This prevents to use the output of a single RAM³S instance to feed multiple RAM³S instances, because each result would only be processed by one of them only. Therefore, the topology of RAM³S instances can only take the form of a pipeline, instead of a more general Directed Acyclic Graph (DAG). As a further extension of RAM³S (which we are already considering), the `Writer`/`Reader` communication protocol could be modified into a Publish/Subscribe model, thus allowing DAG topologies of RAM³S instances.

### 5.2. Adding Samza

In order to include Samza to the set of stream processing frameworks supported by RAM³S, we then need to specify how to exploit objects implementing the RAM³S `Receiver` and `Analyzer` interfaces. Thanks to the generality of the Samza APIs, it is quite straightforward to build a Samza custom stream using the RAM³S `Receiver`. As to the `Analyzer`, this can be directly mapped to the `TaskApplication` Samza object. Due to the compatibility with other frameworks, in RAM³S we decided to exploit the synchronous `StreamTask` Samza interface, whose `process()` operation corresponds to the `analyze()` method in the RAM³S `Analyzer`. Figure 7 shows the UML diagram of classes involved in the creation of a RAM³S application with Samza. In details, the `SamzaApplication` class uses the `ApplicationFactory` to obtain the `Receiver` and the `Analyzer` specific for the application at hand, then obtains the `Reader` and the `Writer` (not shown in the diagram) from the `MessageBrokerFactory`. Finally, a `SamzaTaskApplication` is created that uses a `SamzaStreamTask`, invoking the `analyze()` method to process a single message (i.e., a `MMObject`). We note here that several other service classes and methods (not shown in the diagrams) were required to adapt RAM³S to the Samza APIs. We deliberately chose to hide such intricacies to only characterize elements that are of interest for the developer of MM stream analysis applications, because she needs to write code to interface with them.

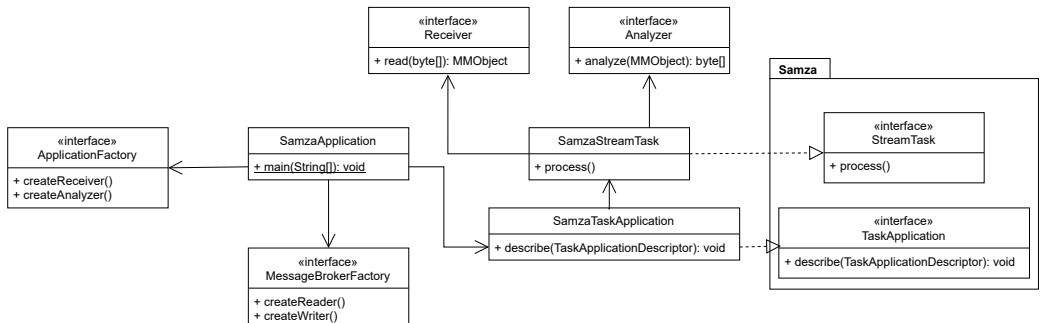

**Figure 7.** UML diagram of the Samza version of a RAM³S application.

The interaction of the different RAM³S objects involved in the processing of a message/`MMObject` is shown in Figure 8 as a UML sequence diagram. Upon invocation of the `process()` method, the `SamzaStreamTask` requests the `Receiver` to read the next message; this is performed by the `Reader` whose result is used to create a new `MMObject`. The just created object is then sent to the `Analyzer` and the result of the `analyze()` method is finally written on the `Writer`.

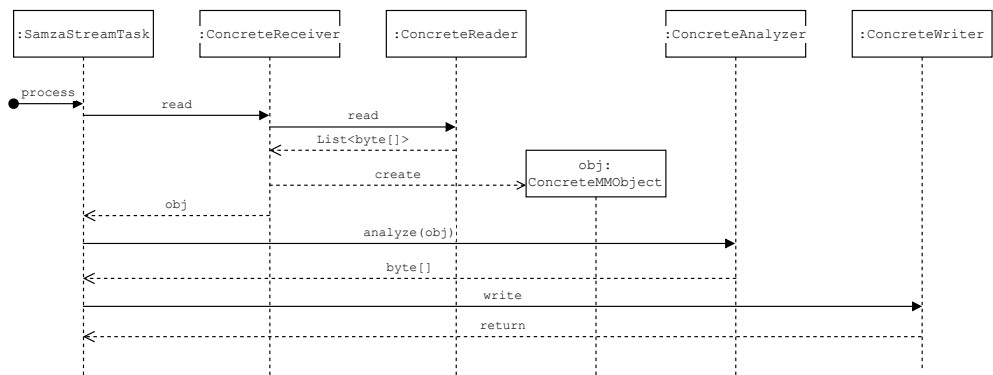

**Figure 8.** UML sequence diagram showing the interaction between objects during the processing of a message in the Samza version of a RAM$^3$S application.

## 6. Experimental Results

The experimental results we present in this section were collected in the *datalab*, (http://www-db.disi.unibo.it/research/datalab, accessed on 3 December 2021) a research lab equipped with a computer cluster using BigData platforms, such as Hadoop, Spark, Storm, Flink, and Samza. Research at datalab focuses on tackling problems that cannot solved by means of traditional techniques, due to the inherent nature of data or to the architecture of the system at hand.

The computing cluster hosted at datalab is composed of 16 PCs with an Intel Core 2 Duo, 2.80 GHz CPU, and 4 GB of RAM, connected with a 100 Mbit/s Ethernet network. An additional identical machine was designated as the master. Finally, an external machine hosts the message broker, which is used to handle input/output streams. Figure 9 depicts the computing environment in our datalab.

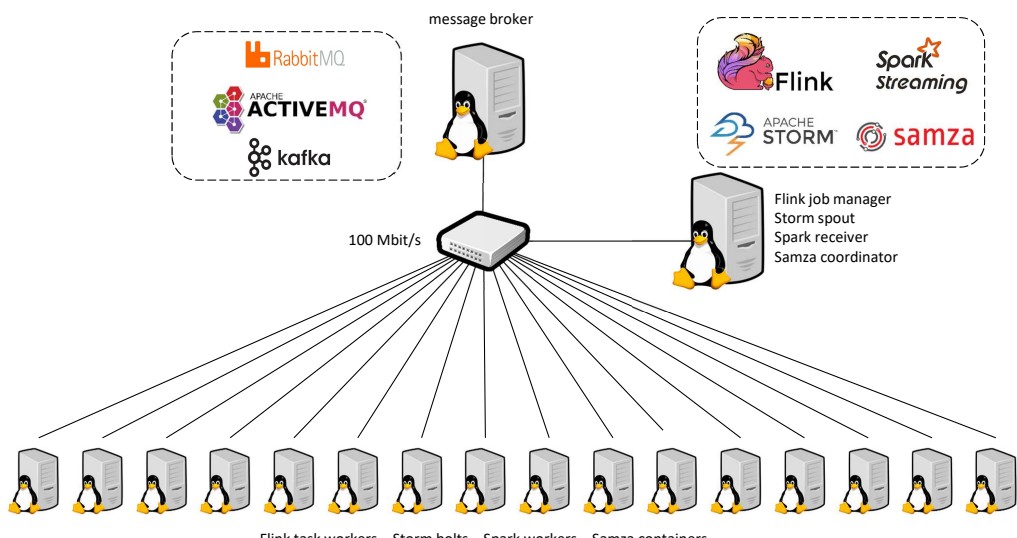

**Figure 9.** The datalab computing environment.

### 6.1. Message Brokers Comparison

To evaluate the performance of message brokers, we performed an experiment to test the processing speed of the brokers, rather than their sending speed, i.e., we compared the speed at which they write and receive a message on and from a topic (in the case of Kafka) or a queue (for RabbitMQ); for ActiveMQ, we considered both its queue and topic modalities. For each message, size and timestamp have been saved upon sending and receiving.

The experiment aims at comparing throughput of the three message brokers when feeding messages with constant size. Table 1 reports the processing speed of RabbitMQ, ActiveMQ (in both the topic and the queue modalities), and Kafka, when feeding messages of constant size to a RAM$^3$S `Receiver` (in this case, the `Analyzer` is empty in order to avoid delaying the message reception). All message brokers exhibit similar performance, with ActiveMQ (in the topic modality) achieving the best result for small messages and RabbitMQ being the winner for larger message sizes. Kafka appears to be a valuable alternative, obtaining good performance in both considered scenarios. For this reason, in the following experiments we will only use Kafka as message broker.

**Table 1.** Performance comparison between the three considered message brokers: Message size 1 KB (a) and 1 MB (b).

| | (a) | (b) |
|---|---|---|
| **Message Broker** | **Throughput (MB/s)** | **Throughput (MB/s)** |
| RabbitMQ | 38.9 | 44.2 |
| ActiveMQ (queue) | 37.5 | 40.5 |
| ActiveMQ (topic) | 43.5 | 43.5 |
| Kafka | 42.8 | 41.1 |

### 6.2. Use Cases

In this section, we describe experiments we performed on the different multimedia applications implemented on top of RAM$^3$S. In all considered use cases, the input multimedia data stream is a video, which is first segmented into individual images, that are then fed to the message broker.

#### 6.2.1. Face Recognition

The first experiment we report considers the use case of face recognition described in Section 1.1: Our real dataset consists of the YouTube Faces Dataset (YTFD) [15], including 3425 videos of 1595 different people, with videos having different resolutions (480 × 360 pixels is the most common size) and a total of 621,126 frames containing at least a face (on average, 181.3 frames/video). The average dimension for an image is 22.3 KB. Results are included in Table 2. As in following experiments, the table reports (for each framework) the obtained throughput with 16 workers in terms of images/MB processed per second.

**Table 2.** Performance comparison between the four considered frameworks for the face recognition use case.

| Framework | Throughput (Images/s) | Throughput (KB/s) |
|---|---|---|
| Spark | 8.1 | 170.1 |
| Storm | 17.2 | 361.2 |
| Flink | 37.7 | 791.7 |
| Samza | 11.4 | 239.4 |

#### 6.2.2. Printed Text Recognition in Videos

The objective of this application, which extracts printed text within video streams, is twofold:

1.  Identifying "critical" videos by analyzing and automatically interpreting the streams of a significant data sample; and
2.  Defining new useful services in the context of monitoring proselytizing phenomena by terrorist groups (this is of utter importance, for example, according to the *Christchurch Call to Action* (https://www.christchurchcall.com/, accessed on 3 December 2021)).

In this case, key frames were extracted from a dataset of about 100 YouTube videos dealing with terrorism and proselytism. A pre-filtering is performed in the `Receiver` to eliminate superfluous frames, maintaining only those in which true information is present. This was obtained with a frame-to-frame analysis, eliminating frames that are too similar to each other. The summarization process exploits the functionalities of the SHIATSU video retrieval tools [16]. When the summary has been obtained, retained images are fed to `Analyzers` which analyze them by:

1. Filtering each image to segment text and logos/symbols from the background;
2. Performing OCR by using the Tesseract library, (https://github.com/tesseract-ocr, accessed on 3 December 2021) and/or extracting logos by using the OpenIMAJ library (http://openimaj.org/, accessed on 3 December 2021);
3. Once the text detection phase is over, extracted text and logos are compared with those included in the knowledge base, so as to recognize the ones that have to be considered critical.

Table 3 reports experiments performed using this specific use case (average image size is about 61 kB).

**Table 3.** Performance comparison between the four considered frameworks for the text recognition use case.

| Framework | Throughput (Images/s) | Throughput (KB/s) |
|---|---|---|
| Spark | 6.5 | 396.5 |
| Storm | 7.2 | 439.2 |
| Flink | 5.4 | 329.4 |
| Samza | 7.4 | 451.4 |

6.2.3. Plate Recognition

Automatic license plate recognition (ALPR) concerns the identification of vehicle license plate from an image or video [17]. Applications of ALPR range from electronic toll collection on pay-per-use roads to speed limit enforcement and traffic control.

To identify a plate in a given image we:

1. Perform plate localization to discover the plate in the image;
2. (Possibly) rotate the image to correct the possible skewing of the plate;
3. Segment characters within the plate to detect them;
4. Recognize the extracted characters using Optical Character Recognition (OCR) techniques;
5. If the recognized plate is included in the knowledge base of "known" plates, the alarm is raised.

Examples of plate recognition are shown in Figure 10.

In our implementation, we exploited the open source C++ library OpenALPR, (http://www.openalpr.com/, accessed on 3 December 2021) that uses OpenCV (https://opencv.org/, accessed on 3 December 2021) for image processing and Tesseract for OCR. Experiments were performed on the OpenALPR benchmark, (https://github.com/openalpr/benchmarks/tree/master/endtoend/, accessed on 3 December 2021) including pictures of cars taken under different light conditions with resolution ranging between $640 \times 480$, $800 \times 600$, and $1792 \times 1312$ pixels.

Table 4 reports experiments performed using this specific use case (average image size is about 340 kB). In this case, an additional test was performed to stress the network, feeding images with size of about 1 MB.

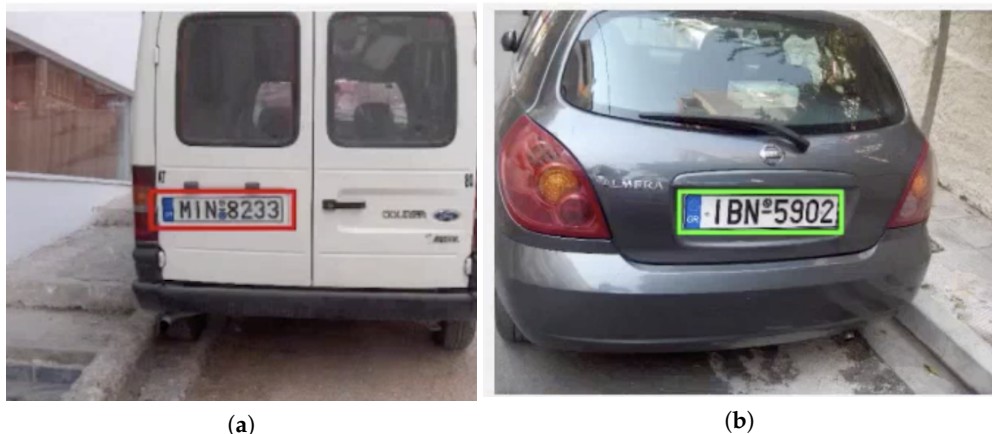

| (a) | (b) |
|-----|-----|

**Figure 10.** Plate recognition use case: known (**a**) and unknown (**b**) plate.

**Table 4.** Performance comparison between the four considered frameworks for the plate recognition use case: normal image size (a) and large image size (b).

| Framework | Throughput (Images/s) | Throughput (KB/s) |
|-----------|----------------------|-------------------|
| Spark | 18.0 | 6120 |
| Storm | 17.3 | 5882 |
| Flink | 15.4 | 5236 |
| Samza | 18.4 | 6256 |
| (a) | | |

| Framework | Throughput (Images/s) | Throughput (KB/s) |
|-----------|----------------------|-------------------|
| Spark | 5.2 | 5183 |
| Storm | 5.2 | 5234 |
| Flink | 7.4 | 7412 |
| Samza | 6.2 | 6231 |
| (b) | | |

## 6.3. RAM$^3$S vs. Samza

Our final experiments aim to compare RAM$^3$S with Samza. Indeed, the simplified interface offered by Samza to the developer (with respect to those of the other considered frameworks) make it a competitor of RAM$^3$S, when it comes to consider abstracting the developer from the details of the underlying stream processing platform. It is thus desirable to compare the performance of RAM$^3$S and Samza both in ease of use and in throughput.

In the first experiment, we compare the number of code lines required to implement the face recognition use case (see Section 1.1) when using the different considered frameworks. As results in Table 5 show, Samza is very effective in reducing the developer burden, requiring the minimum number of code lines. However, this result is only valid when Samza is used in connection with the Kafka message broker: When using a custom connector, to abstract from the used message broker, the performance of Samza is the worst one. On the other hand, RAM$^3$S is the second best, but is already independent of the used message broker, thus representing the most effective general framework.

**Table 5.** Ease of use in implementing the face recognition use case over the considered frameworks.

| Framework | Lines of Code |
| --- | --- |
| Spark | ~170 |
| Storm | ~150 |
| Flink | ~200 |
| Samza (Kafka) | ~50 |
| Samza (custom connector) | ~250 |
| RAM$^3$S | ~120 |

In the second experiment, we compared the throughput of using Samza alone or within RAM$^3$S, again for the face recognition use case. As reported in Section 1.1, the throughput obtained when using RAM$^3$S on top of Samza is about 240 KB/s. The use of Samza alone using the custom connector allows to reach the throughput of about 280 KB/s, with a 15% improvement, while using a Kafka connector we obtain a further speed increase, reaching 320 KB/s, with a 35% improvement over the original case.

The lesson we learned from this last experiment was twofold:

1. The introduction of a custom connector to manage the input stream is slowing down the overall system;
2. The tight coupling between Samza and Kafka results in high performance (see Section 4.3).

The system slowdown is clearly caused by this overlapping of levels: This is inevitable if we want to achieve generality in using any stream processing framework coupled with any message broker.

## 7. Conclusions

The first result obtained by RAM$^3$S is the simplified implementation of complex multimedia stream analysis applications, by resulting in a reduction in code lines one has to write to run them in a distributed computing environment. This clearly helps developers in quickly scale out their multimedia stream data mining applications which were originally conceived for a non-distributed scenario. This benefits an easier extraction of value and knowledge from massive multimedia data streams.

A second advantage is the independence of the implementation from the underlying exploited Big Data platform, since RAM$^3$S can indifferently run on top of Spark, Storm, Flink, or Samza, and the decision can be performed at runtime. This allows, for example, to devise which, of the four platforms, is the most suitable for the scenario at hand, considering both the characteristics of available hardware and the requirements of the online application (e.g., limited latency, scalability, high throughput). Indeed, as recently acknowledged [18], the benchmarking of stream engines has become quite a battleground, as contrasting results from Spark [19] and Flink [20] have been published. Quoting from [18]:

> *"It is better not to believe benchmarking these days because even a small tweaking can completely change the numbers. Nothing is better than trying and testing ourselves before deciding."*

However, implementing even a simple application on top of different stream processing platforms could be quite hard, since the developer has to deal with details specific to each framework. Here, the generality and wide applicability of RAM$^3$S are of immediate help, since the same code could be executed on top of any of the four included streaming platforms. A relevant step in this direction is being pursued by Google, whose Apache Beam platform (https://beam.apache.org/, accessed on 3 December 2021) has the goal of writing stream (and also batch) processing applications independent of the underlying engine (called *runner*), so that the resulting code could be executed on any of the supported runners (which also include the proprietary Google Cloud Dataflow (https://cloud.google.com/dataflow/, accessed on 3 December 2021) [21].

To conclude, we would like to point out that, since the scenario of stream processing platforms is evolving at a breathtaking speed (an example is the short life of Apache Apex (https://apex.apache.org/, accessed on 3 December 2021)), we plan to continue in metamorphosing RAM³S, e.g., to include other streaming engines. For example, we are considering including Apache Kafka not only as a message broker, rather as a first class citizen of the streaming world, thanks to its Kafka Streams API. A further direction for future work is the deployment modality of underlying platforms: At the moment every streaming engine is deployed in stand-alone mode, while including a resource manager such as Apache Hadoop Yet Another Resource Negotiator (YARN) could automatically add fault-tolerance and resource balancing to the overall architecture. Finally, a further extension of the `Reader` and `Writer` classes would allow to connect more than a `Reader` to any `Writer` (and vice versa) so that the whole multimedia stream processing architecture can assume the form of a Directed Acyclic Graph.

**Author Contributions:** Both authors equally contributed to prepare the manuscript. All authors have read and agreed to the published version of the manuscript.

**Funding:** This research received no external funding.

**Institutional Review Board Statement:** Not applicable.

**Informed Consent Statement:** Not applicable.

**Data Availability Statement:** Data presented in the paper are available from the authors upon request.

**Acknowledgments:** The authors acknowledge the students of datalab for implementation of RAM³S and specific use cases. For the title and opening sentence, the authors thank the Kafka Project, and Ian Johnston in particular. (http://www.kafka.org/index.php?aid=170, accessed on 3 December 2021).

**Conflicts of Interest:** The authors declare no conflict of interest.

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
