# Peer review of "The Metamorphosis (of RAM3S)"

_applsci, doi:10.3390/app112411584_

Round 1

Reviewer 1 Report

Research is interesting with adequate use of modern Big Data technologies. It will be good to update some parts of research:

  • Authors should extend method chapter with adequate UML contents diagram and diagram where they will show elements of the system.
  • Also it will be good to show adequate hdfs document which is part of the process. Just show the structure of that document.
  • Also, if there is developed User Defined Types or User defined Functions, it should be mentioned and presented in the text.

Reviewer 2 Report

This work extends the RAM3S framework which is used for the analysis of multimedia streams. RAM3S, which was presented by the authors in previous publications, aims to define a middleware between multimedia stream analysis techniques and Big Data streaming platforms, allowing the analysts and developers to set up multimedia stream analysis applications over different Streaming/BigData technologies (like Spark Streaming, Storm, Flink). In this work, the authors extend the framework in the following manner. Initially, the authors generalize the framework in order to support different message brokers (e.g., Kafka); note that the previous version of the framework used only RabbitMQ. Using this extension, pipelines of processing steps can be defined. In addition, the authors extend the RAM3S framework to support Apache Samza as an additional processing engine. Finally, a set of experiments is presented for 3 use cases.

Although the paper is well-structured and easy to read, the contributions of this work are limited to small adjustments in order to support additional message brokers (in the initial version of the framework, it seems that the implementation was tightly bound into RabbitMQ); note that the main contributions cover a single page into the paper. Moreover, the integration with Samza seems to be straightforward. As for the cascading operations, it is not clear whether these operations can be used to define a DAG (e.g., multiple steps output MM objects into the same stream which can then be used as an input to another step). 

Reviewer 3 Report

The work is a follow up paper describing improvements over a previously published framework for streams (in particular for multimedia data streams) called RAMS3S.
RAMS3S can be seen as a middleware that allows the use of different stream processing systems under the same API, simplifying the development of an otherwise complex hardcoded combination of software components. Different real use cases are shown, regarding human face and driving license recognition.

In the new version, the framework has been extended to support:
1) a variety of message brokers (that is, software modules in charge priority queues for the data to be dispatched to working nodes), while previous version only supported the RabbitMQ message broker.
2) Apache Samza: while the previous version already supported a number stream processing systems, Apache Samza has been added to the list.

An important contribution of the paper, other than technically describing the new features, is the experimental section. It compares performance of different message brokers and different stream processors, thanks to the wide choices available on RAMS3S, that also acts as an handy platform for testing purposes.
Finally, RAM3S is appropriately compared with Apache Samza directly, under different configurations, showing an expected overhead in terms of throughput, which seems to be affordable in order to the flexibility offered by RAMS3S.

Round 2

Reviewer 2 Report

The authors provided more details regarding the support of Samza, as well as, they enhanced the analysis of the framework.

As a minor comment, in lines 426 and 429, the text exceeds the page width.